# Brief Communication: Thinning of debris-covered and debris-free glaciers in a warming climate

Argha Banerjee

Earth and Climate Science, Indian Institute of Science Education and Research, Pune, India

*Correspondence to:* Argha Banerjee (argha@iiserpune.ac.in)

**Abstract.** Recent geodetic mass-balance measurements reveal similar thinning rates on glaciers with or without debris cover in the Himalaya-Karakoram region. This comes as a surprise as a thick debris cover reduces the surface melting significantly due to its insulating effects. Here we present arguments, supported by results from numerical flowline model simulations of idealised glaciers, that a competition between the changes in the surface mass-balance forcing and that of the emergence/submergence velocities can lead to similar thinning rates on these two types of glaciers. As the climate starts warming, the thinning rate on a debris-covered glacier is initially smaller than that on a similar debris-free glacier. Subsequently the rate on the debris-covered glacier becomes comparable to and then larger than that on the debris-free one. The time evolution of glacier-averaged thinning rates after an initial warming is strongly controlled by the time-variation of the corresponding emergence velocity profile.

## 1 Introduction

A knowledge-gap related to debris-covered glacier dynamics affects our understanding of the past and future of Himalayan glaciers in a changing climate (Scherler et al, 2011). The supra-glacial debris cover present over the ablation zone of a glacier induces qualitative changes in its dynamic response (Naito et al, 2000; Vacco et al, 2010; Benn et al, 2012; Banerjee and Shankar, 2013; Anderson and Anderson, 2015) due to a suppressed melt-rate under a thick debris layer (Nakawo and Young, 1982; Mattson et al, 1993). Whereas a thin debris cover is expected to accelerate melt due to its low albedo. While responding to a warming climate, debris-covered glaciers exhibit a larger climate sensitivity, a longer response time (Banerjee and Shankar, 2013), a decoupling of volume and length changes and the formation of a slow-flowing stagnant downwasting tongue (Scherler et al, 2011; Banerjee and Shankar, 2013). Despite several efforts to model and understand the dynamics of debris-covered glaciers with various degrees of sophistication (Naito et al, 2000; Vacco et al, 2010; Banerjee and Shankar, 2013; Anderson and Anderson, 2015; Rowan et al, 2015), challenges still remain. This task is made more difficult by our limited understanding of the time-evolution of the debris extent (Anderson and Anderson, 2015), the variability of debris thickness, and common occurrences of highly dynamic supraglacial ponds and ice-cliffs that cause intense localised melting (Sakai et al, 2000; Miles et al, 2015; Steiner et al, 2015).

A curious fact that has emerged from large scale remote sensing measurements of glaciers in the Himalaya and Karakoram during the first decade of 21st century is a similar magnitude of thinning of glacial ice irrespective of the presence of

supraglacial debris-cover (Kääb et al, 2012; Gardelle et al, 2012; Nuimura et al, 2012; Gardelle et al, 2013) and this may seem

counter-intuitive. A thick debris cover, due to its insulating properties, significantly inhibits the melt of the underlying ice - so

much so that in the debris-covered part of the glacier, the specific melt-rate does not increase with decreasing elevation. Rather,

it saturates to some lower bound or even decreases downglacier (Banerjee and Azam, 2015). On the other hand, on a debris-

free glacier the melt rate typically increases monotonically as elevation decreases. Why then should both the glacier-types

experience similar rates of thinning as climate warms up?

Heuristic arguments were offered by various authors to reconcile with this apparent paradox. Kääb et al (2012) suggested that

the insulating effect of the debris cover might be compensated for at the scale of the whole ablation zone due to an enhanced

melting from the thermoskarst processes, namely supra-glacial ponds and ice-cliffs that are often present on the debris-covered

glaciers. These features, due to an associated discontinuous debris cover, experience large localised melting. Given that these

features typically contribute $\sim 10 - 20\%$ of the total melt (Sakai et al, (2000); Reid and Brock, (2014)), it is unlikely that

they can lower the glacier-wide mean melt rate on debris-covered glaciers sufficiently so that it matches that on the debris-free

glaciers. Field measurements by Vincent et al (2016) seem to confirm this. It was also conjectured that a reduction in ice-flux

from upstream areas to the stagnant tongue may be behind the larger-than-expected thinning of debris-covered glacial ice (Kääb

et al, 2012; Gardelle et al, 2012). Nuimura et al (2012) too pointed out the possible role of reduced flux into the low-slope

slow-moving stagnant tongues of large debris-covered glaciers. However, a quantification of this flux-effect is missing as yet.

On the other hand, Banerjee and Shankar (2013) showed that a reduced melt-rate on a debris-covered glacier does not affect

the volume response of the glacier qualitatively, in stark contrast with its drastic effect on the length response of the glacier.

However, their model results (figure 3d of Banerjee and Shankar (2013)) show a relatively larger thinning rate on the debris-free

glaciers in response to a rapid warming. Also, it was reported that in the Pamir-Karakoram-Himalaya, depending on the region

chosen, geodetic measurement yielded decadal thinning rates of debris-covered ice that were either larger or smaller than, or

similar to that of debris-free ice (Gardelle et al, 2013). The present scenario is summed up neatly by Vincent et al (2016), "This

question of area-averaged melting rates over debris-covered or clean glacier ablation areas remains unanswered".

In this contribution, we analyse the rate of thinning on debris-covered and debris-free glaciers in a warming climate using

a one-dimensional flowline model of idealised glaciers (Banerjee and Shankar, 2013; Banerjee and Azam, 2015). We conduct

simple numerical experiments to investigate the role of the magnitude of warming rate, the ice dynamics (i.e. the changes in the

flux-gradient profiles or equivalently that in emergence/submergence velocities), and that of the surface mass balance forcing,

in controlling the thinning rates on these two glacier types.

## 2   Glacier response to instantaneous warming

An easy-to-analyse piece of this problem is the behaviour of a steady-state debris-covered or debris-free glacier immediately

after an instantaneous rise of temperature (or equivalently that of the equilibrium line altitude (ELA)). In a steady state, the ice-

thickness profile remains constant due to a stable balance between the surface ablation (accumulation) rate and the emergence

(submergence) velocities. Dictated by mass conservation of incompressible ice, the emergence or submergence rate equals the

negative gradient of the flux, $F(x)$. After an instantaneous change in ELA, the surface mass balance values change, but the

viscous ice flow takes a characteristic longer time to relax. Therefore, the local thinning rate is initially just the difference in

specific mass balance, $B(x)$, before and after the change in temperature. However this is valid only over a time scale that is

short compared to the flow-relaxation time.

Let us consider two idealised model glaciers. Glacier A does not have any debris cover and has a linear mass-balance profile.

Glacier B has a supraglacial debris cover on its lower ablation zone where the ablation rate saturates to a value of -2 m/yr

(figure 1b). This idealised mass-balance profile for the debris covered glacier is motivated by data from Himalayan glacier

(Banerjee and Azam, 2015). Similar simplified mass-balance profiles have been used to analyse the response of the debris-

covered Himalayan glaciers (Banerjee and Shankar, 2013; Banerjee and Azam, 2015). In a real glacier, the possible variability

of debris thickness and ephemeral thermokarst features (ponds and ice-cliffs) cause significant spatial variation of the melt-rate

in the debris-covered parts of the glacier. However, a relatively fast advection of these surface features would imply that a

long-term mean melt-rate at a specific location is a well defined quantity. This justifies the simplified mass-balance profile

employed here. Further, the observed thinning rate values in the Himalaya are obtained for a large set of glaciers so that the

possible effects of specific details of mass-balance profile of individual glaciers would be averaged out.

In figure 1a, 1b we show mass-balance profiles for the idealised model glaciers before and after an instantaneous rise of

ELA, $\Delta E =$50 m. It is assumed here that the mass-balance shape remains the same and changes only by a shift of ELA. In

practice the debris layer may thicken and the debris-covered area may grow in a warming climate, affecting the shape of the

melt-rate profile. However, it is known that above a debris thickness of $\sim$ 10 cm, the decrease in melt-rate with a thickening

debris layer is small (Juene et al, 2014). Therefore such changes can safely be neglected as a first approximation. The possible

changes in supraglacial ponds/ice-cliffs are neglected at this level of approximation due to a relatively smaller contribution of

these features to the total melt, as discussed before. This assumption of an invariant shape allows for the possible increase in

debris extent with warming as the upper boundary of the region with saturated melt-rate moves up with the ELA. Overall these

simplifications allow us to focus on the role of ice-flow dynamics in controlling the downwasting of glaciers in a warming

climate.

As is clear from figure 1a, glacier A responds initially with a uniform glacier-wide thinning rate, $\langle \frac{dh}{dt} \rangle_A = \beta \Delta E$, right after

the ELA change. Here $\beta$ is the mass-balance gradient. For glacier B, a uniform thinning operates only on the debris-free upper

part of the glacier and the lower part has not thinned at all (figure 1b). Thus, glacier B has a lower mean thinning rate to start

with that is given by $\langle \frac{dh}{dt} \rangle_B = (1 - f_d)\beta \Delta E$, where $f_d$ is the debris-covered fraction. Remarkably these expressions do not

involve the length of the glaciers. Also, the initial lack of thinning on the debris-covered glacier is independent of the actual

value of the melt-rate under the thick debris layer (assumed to be 2 m/yr here) and depends only on the general shape of the

melt-curve (figure 1b).

A more general mass-balance profile for a debris-covered glacier than the one considered above would involve a smaller or

inverted mass-balance gradient in the debris-covered parts (Banerjee and Azam, 2015). Even then, the mean initial thinning

rate on such a glacier would be less than that of a corresponding debris-free one. This delayed thinning of the debris-covered

terminus is consistent with the formation of a slow-flowing stagnant tongue with very little retreat as observed on debris-

covered glaciers in the Himalaya-Karakoram (Scherler et al, 2011). This raises confidence in the minimal description of such glaciers that is being used here. In case of an inverted mass-balance, a transient thickening of the lower ablation zone is observed, though this is likely to be an artifact of the assumed fixed shape of the mass-balance curve.

Thus, a debris-covered glacier starts with a lower value of mean thinning rate compared to a debris-free one (as $\langle \frac{dh}{dt} \rangle_A >$ $\langle \frac{dh}{dt} \rangle_B$). The ice fluxes then respond to the mass-balance change and subsequent evolution of the flux-gradient profile or equivalently that of the emergence velocity profile alters the distribution and magnitude of the thinning rate. Though the detailed spatial and temporal pattern of such changes are difficult to predict, at some later stage the thinning rate on glacier B is likely to become larger than that on glacier A. This is because, 1) the debris covered glacier B has a larger climate sensitivity (Banerjee and Shankar, 2013) as compared to glacier A and thus loses more mass for a same change in the ELA; 2) On glacier B, the lower ablation zone responds to the perturbation with a delay. There must be an intermediate crossover period as well, where the thinning rates on both the glaciers have similar magnitude within measurement errors.

## 3  Numerical investigations

To verify above claims on the nature of the evolution of thinning rate on glacier A and B, we perform a set of numerical experiments with 1-d flowline models of glacier A and B. The model glaciers have bedrock slope of 0.1 and mass balance gradient $\beta = 0.007$ yr$^{-1}$. See Banerjee and Shankar (2013) for further details of the flowline model used. Note that these glaciers are identical above the debris-covered region (figure 1a, 1b). The initial steady-states are prepared by running the models with a fixed value of ELA for 500 (900) years for glacier A (B). The steady-state length of the simulated glaciers are in the range 6–14 km. Subsequently, the following ELA perturbations are switched on at $t = 0$:

1. An instantaneous rise by 50 m.

2. A total rise of 50 m in steps of 5 m every five years.

3. A total rise of 30 m in steps of 1 m every five years.

In all the three experiments the net warming is similar, but rates and durations of the ELA perturbations different (1. an instantaneous warming; 2. a rate of 10 m/decade for 50 years; 3. a rate of 2 m/decade for 150 years). In experiment (3), we restrict the total ELA rise to 30 m so as to limit the duration of the experiment to 150 years to facilitate comparison with the other two experiments.

### 3.1  Results and discussions

### 3.1.1  Initial thinning rates

Just as argued in section 2, mean thinning rate profiles obtained after a year in experiment (1) show uniform thinning all over glacier A and in the upper part of glacier B (figure 1e, 1f). In contrast, the debris-covered parts of glacier B show no thinning. At this point, the flux gradient profile (same as the negative of emergence velocity), $\frac{dF}{dx}$, has not changed significantly from the

initial steady mass balance profile $B(x)$ (figure 1c, 1d). Further, the initial thinning rates for glaciers A and B in experiment (1) are quite accurately given by $\beta \Delta E$ (0.35 m/yr) and $(1 - f_d)\beta \Delta E$ (0.22 m/yr) respectively. All these results are consistent with our arguments as outlined in section 2. The thinning rate trends for finite warming rates follow a similar pattern, with the difference between two rates during the initial phase growing for larger value of the warming rate (figure 2; experiments (2) and (3)).

### 3.1.2 Time evolution of the thinning rates

A thinning of ice in the ablation zone takes place when the local melt-rate overcomes the local emergence velocity. Data from experiment (1) show that the initial profile of the thinning rate gets modified at later times largely due to a changing profile of $\frac{dF}{dx}$ (figure 1e, 1f). After the initial rapid change, the competing term of mass balance rate varies weakly with time - due to a feedback from the changing ice-thickness. Therefore, the evolution of the spatial distribution and the mean value of the thinning rate are mostly *dynamically* controlled by a changing emergence velocity profile. While this is in general true for both the glaciers types, emergence velocity profile on the lower ablation zone of the debris-covered glacier shows a *delayed* response (figure 1f) which leads to a low glacier-averaged initial thinning rate for these glaciers.

Consistent with arguments given in section 2, the mean thinning rate on glacier B has a lower magnitude initially. Subsequently the thinning rate matches and then overtakes that on glacier A (figure 2). This illustrates that depending on the stage of response, a debris-covered glacier can have a smaller, larger or similar mean thinning rate as compared to that on a corresponding debris-free glacier. As expected, similar trends are obtained in experiments with finite warming rates. However, at the limit of a very low rate of warming, the differences between the thinning rates on the two glaciers are small (figure 2; experiment(3)). The cross-over time seems to be controlled by the rate of warming.

While we have considered the glacier-wide thinning rate, the same conclusions are obtained if one compares only the lower part of the two glaciers as they are identical in their upper parts. The thinning rate when measured on a regional scale, is an average over glaciers having different size, shape, bedrock-profile, and even history of warming. Clearly, in the light of the above discussion, this may lead to larger, smaller or similar mean thinning rates in the the debris-covered glaciers as compared to the debris-free glaciers from the same region, in agreement with observations by Gardelle et al (2013).

## 4 Conclusions

We provide very general arguments that debris-covered glaciers, while responding to a warming climate, can have smaller, larger or similar thinning rates as compared to corresponding debris-free glaciers. Thinning of glaciers is controlled by a competition between a changing mass-balance and the emergence velocity profile. A debris-covered glacier starts with a smaller glacier-averaged thinning rate, but overtakes that of a debris-free glacier at later stages of evolution. The initial difference in the corresponding thinning rates depend on the balance gradient and the debris-covered fraction. The changes in local melt-rates control the thinning of glacial ice immediately after an instantaneous warming, whereas a stronger variation of the

corresponding emergence-velocity profile dictates the evolution of the thinning of ice at subsequent stages. Our arguments are validated against results from flowline model simulations of idealised glaciers.

*Acknowledgements.* This work is supported by DST-SERB grant no SB.DGH-71.2013 and DST-INSPIRE Faculty award (IFA-12-EAS-04).

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

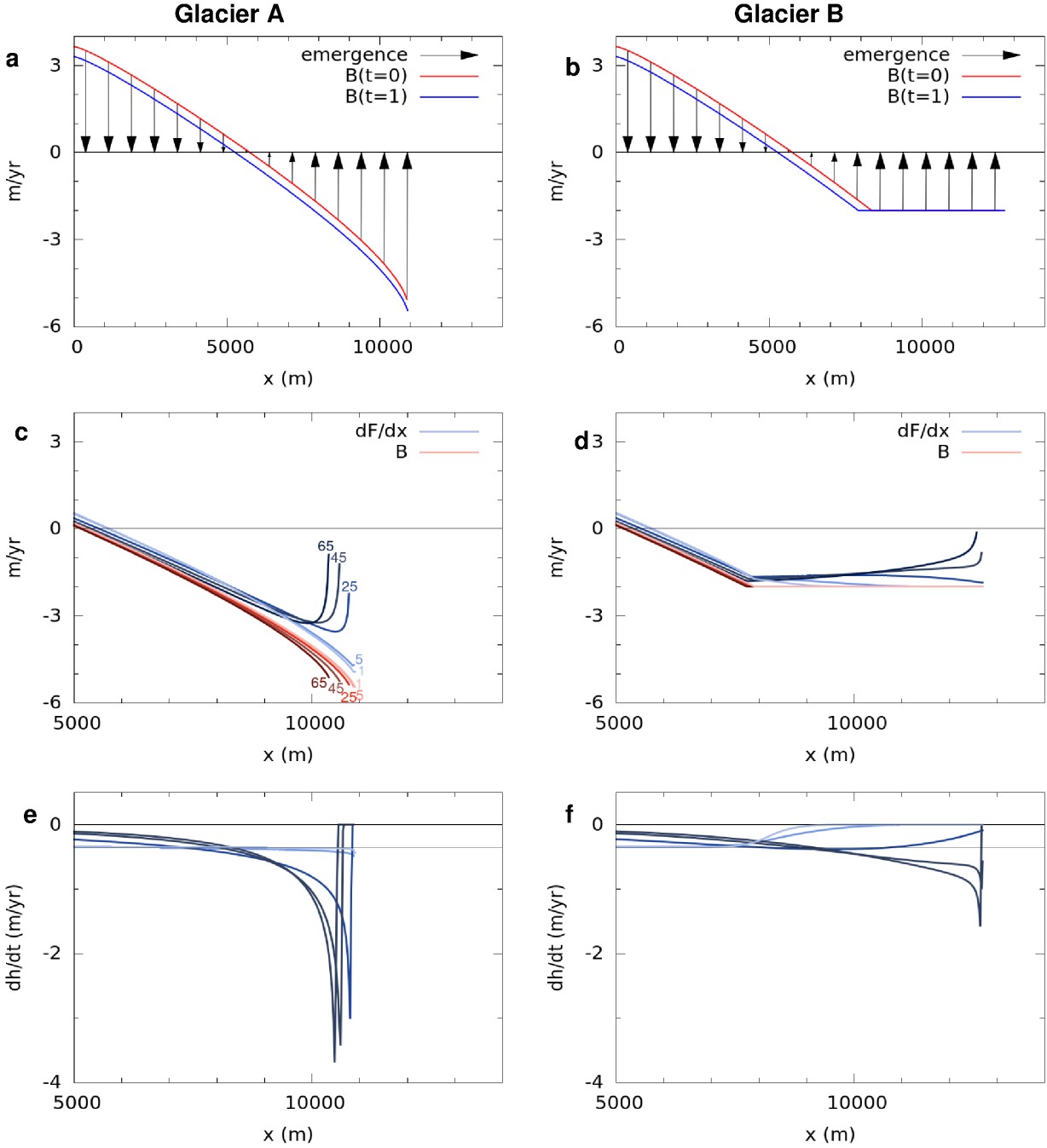

**Figure 1.** (a, b) The specific mass-balance as a function of position for the initial steady-states of glacier A and B (red lines). Black arrows denote the emergence velocities that balances the surface mass balance at $t = 0$. The blue lines are the surface mass-balance profiles a year after a step change in ELA by 50m (Experiment (1)). (c, d) The specific mass-balance (red lines) and flux-gradient (blue lines) profiles after 1, 5, 25, 45, and 65 years. In (c) the curves are labeled with the corresponding year. (e, f) The thinning rate profiles after 1, 5, 25, 45, and 65 years. Note the horizontal black thin lines at $\beta \Delta E = 0.35$ m/yr (see text for details).

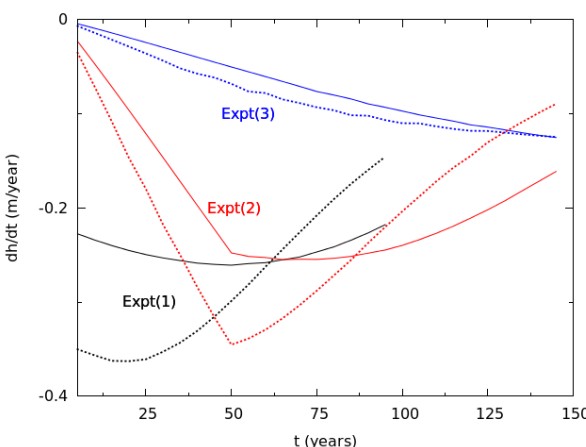

**Figure 2.** Evolution of thinning rates after ELA perturbations are applied to a model debris-covered glacier (solid line) and a debris-free glacier (dotted line). The warming rate profiles for each of the experiments are described in section 3.