# Peer review of "Brief Communication: Thinning of debris-covered and debris-free glaciers in a warming climate"

_The Cryosphere, 2016_

## Referee Comment (RC1) · Anonymous Referee #1 · 17 Aug 2016

This paper presents numerical experiments on response of debris-covered and debris-free glaciers to changing ELA. Different mass balance profiles were assumed for debris-covered and debris-free glaciers to investigate the thinning rate and its temporal evolution. The motivation of the study is observations in Himalaya, in which thinning rates of debris-covered glaciers are not always smaller than those of debris-free glaciers. This is counterintuitive because a debris layer reduces ice melt, which should results in less negative surface mass balance. Based on a series of simple experiments, the authors showed that debris-covered glaciers may thin more rapidly than debris-free glaciers under a warming climate. This happens because vertical straining due to ice flow plays a role in glacier thickness change, and such dynamic response of debris-covered glaciers is slower than that of debris free glaciers.

It is difficult to find the significance of the study. Glacier thinning occurs by a com-

bination of the surface mass balance and the emergence velocity. Initial change in ice thickness is controlled by surface mass balance, and then affected by changes in glacier dynamics later. Response time of a debris-covered glacier is generally longer than that of debris-free glaciers. All these were frequently argued and well demonstrated in previous studies. Therefore, it is not surprising to see the results shown in Figure 2. Moreover, the model and experimental conditions are very simple (1D flow line model, simple ice dynamics and mass balance). Among others, this study neglects important aspects of a debris-covered glacier, which are listed in the introduction of the paper (line 19-20); time-evolution of the debris extent, variability of debris thickness, and highly dynamic supraglacial ponds and ice cliffs. In any case, the paper is too short to report complex behavior of debris-covered glaciers.

Because of the reasons above, I do not think the paper is worth published in The Cryosphere as "Brief communication", which should be a timely report on new developments, significant advances and novel aspects of experimental and technical methods and techniques. I list below specific comments on the manuscript.

page 1, line19-20: These are very important aspects, but completely neglected in the study.

page 2, line 28: "vertical ablation" is odd. Do you mean "surface ablation"?

page 3, line 3-4: "mass balance shape remains the same" » This is a very crude assumption because the debris layer thickens and lakes are formed.

page 3, line 13-14: The result is not "interesting" if "this is an artifact".

page 3, line 25: What is the unit of the mass balance gradient?

page 3, line 3: Why 30 m (not 50 m)?

page 4, line 4-10: These results are easily expected before the experiments. The results are like that, simply because of the assumptions given to the mass balance.

[Figure]

page 4, line 18: "overtakes that of glacier B" » "glacier A"?

---

## Author Comment (AC1) · 22 Aug 2016

Authors response to **Review comments on "Thinning of debris-covered and debris-free glaciers in a warming climate" by A. Banerjee', Anonymous Referee #1**

I thank the Anonymous referee #1 for a critical review of the discussion paper. In particular, I am glad that the reviewer agrees with the result of the paper. Below I have outlined my arguments against his conclusions regarding inadequacy of the model and lack of significance of the results . The red lines are from the referee's review and corresponding replies are in black.

It is difficult to find the significance of the study. Glacier thinning occurs by a combination of the surface mass balance and the emergence velocity. Initial change in ice thickness is controlled by surface mass balance, and then affected by changes in glacier dynamics later. Response time of a debris-covered glacier is generally longer than that of debris-free glaciers. All these were frequently argued and well demonstrated in previous studies.Therefore, it is not surprising to see the results shown in Figure 2.

Undoubtedly glacier thinning has to be controlled by conservation of mass, a slow dynamics of ice and a fast changing mass-balance forcing. I do not claim to have introduced this ideas here in this paper for the first time.
However, to the best of my knowledge, these basic principles were not applied so far in interpreting the recent large scale thinning data from debris-covered and debris-free glaciers in the Himalaya (Kääb et al , 2012; Gardelle et al , 2012; Nuimura et al , 2012; Gardelle et al , 2013, Vincent et al, 2016), leading to the apparent puzzle that has been outlined in detail in the introduction section (from page1, line 22 to page2, line 19). Even with this long list of well-known papers that have dealt with this issue, Vincent et al (2016) has stated: "This question of area-averaged melting rates over debris-covered or clean glacier ablation areas remains unanswered". This is contradictory to the reviewer's claim and shows that a clear understanding of this effect has been lacking in the present literature so far.
This paper attempts provide a very simple solution to this specific issue from first principles.
If the effect has already been clearly explained in some reference that is not known to me, I am ready to accept that the present contribution is redundant.

Moreover, the model and experimental conditions are very simple (1D flow
line model, simple ice dynamics and mass balance). Among others, this study neglects
important aspects of a debris-covered glacier, which are listed in the introduction of the
paper (line 19-20); time-evolution of the debris extent, variability of debris thickness,
and highly dynamic supraglacial ponds and ice cliffs.

I apologise to the reviewer and the readers for not providing a detailed justification of the simple model used in this paper. I thank the reviewer for pointing this weakness out. Such a discussion would surely be included in possible revised version of this article.

The basic point here is that the relatively fast spatio-temporal variations of melt-rate due to the advecting ephemeral thermokarst features (ponds and cliffs) on the glacier surface and an inhomogeneous debris layer, in combination with a slow response of debris-covered glaciers, imply that avarage melt-rate is a rather well-defined quantity and that is what that controls the thinning dynamics at any given point, $x$, over deacal scale. Moreover, as pointed out in the article, the present data suggest, the theromokarst feature play a relatively weaker role in terms of the total melt - at the level of 10-20% (Sakai et al, (2000); Reid and Brock, (2014)).

In addition, since the quoted thinning data are from a large ensemble of glaciers, another level of averaging over such a large ensemble would get rid of the effects of specific details of the mass

balance the individual glaciers.

Therefore, it is justified to use a simple (and thus tractable) average mass balance curve to investigate the question of large scale thinning rates in glaciers in the Himalaya. The specific melt-curve used here is motivated by data from Himalayan glaciers (Chhota Shigri, Hamtah, Dokriani and Chora Bari glaciers; eg Banerjee and Azam, 2015). A more complicated representative melt-curve would not change our basic results.

There is a possibility that climatic forcing may increase the average melt rate or may lead to higher abundance of ponds/cliffs (discussed later in the reply), and thus changing the mean melt-rates near the tongue. Given the lack of long term data, this effect is hard to quantify at present. The fact that there are number of debris-covered glaciers with large stagnant tongues in the Himalaya (Scherler et al , 2011), may be a pointer that this increase is not very significant in terms of its magnitude. And the idealised mass balance used here, captures the formation of the stagnant tongue quite well.

Notably, the uppper elevation range of the thickly debris-covered region has been assumed to increase in our idealised debris-covered glacier model by the same amount as the ELA, to take care of the possible increase of debris covered area in a simple way.

 In any case, the paper is too short to report complex behavior of debris-covered glaciers.

As explained above, the aim here is to investigate the specific question of decadal scale data of thinning rates of a large collection of debris covered vs debris free glaciers. I do not intend to explain all aspects of the complex behaviour of debris covered glaciers. I believe that the model/paper is adequate for this specific purpose. The complexities alluded to above, would only be relevant in answering more detailed questions like the pattern of thinning in a given glacier and therefore can be safely ignored in the present context.

The existing detailed models, in fact, may not be scalable to simulate a large ensemble of glaciers. Most of these models require high resolution baseline glacier and climate data, which may not be available.

I list below specific comments on the manuscript.
page 1, line19-20: These are very important aspects, but completely neglected in the study.

I have already explained my view on this issue in the response detailed above. These arguments/discussion would be incorporated in possible revised version of the article.

page 2, line 28: "vertical ablation" is odd. Do you mean "surface ablation"?

It would be corrected.

page 3, line 3-4: "mass balance shape remains the same" » This is a very crude assumption because the debris layer thickens and lakes are formed.

A thickening debris layer would affect the mass balance values for sure. However, in the thickly debris covered parts of the glacier this effect would be relatively unimportant. This is evident from the known variation of melt rate under a debris layer (Ostrem curve) that shows smaller decrease in melt-rate in the thick debris limit (more than about ~10 cm). (eg Juen et al, The Cryosphere, 8, 377–386, 2014). And, the possible increase in debris-extent is included in an empirical manner by moving up the saturated portion of the melt-rate curve as ELA goes up (expected in case of melt-out

debris).

On the other hand, supra glacial lakes, as pointed out before, only contribute ~10-20% of the total melt (Sakai et al, (2000); Reid and Brock, (2014)) for specific glaciers studied. Also large-scale studies (eg Gardelle et al, Global and Planetary Change, Elsevier, 2011, 75 (1-2), pp.47-55) reveal that the *supraglacial* lake area is typically only a fraction of a percent of the total glacierised area in the region, and that the total supraglacial lake area is growing at a rate of a few 10's of percent or less per decade (with large uncertainties in the estimates). So the net effective lowering of melt rates due these possibly increasing supraglacial lakes can be ignored in the first approximation.

These discussions would be incorporated in the revised draft.

page 3, line 13-14: The result is not "interesting" if "this is an artifact".
I agree with the reviewer and appropriate changes would made.

page 3, line 25: What is the unit of the mass balance gradient?
Units would be specified.

page 3, line 3: Why 30 m (not 50 m)?
A change of 50m at the rate of 1m every five year, requires a total of 250 years, stretching the time axis of the figure 2 – that is why we had truncated it at 150 years ie a total change of 30 years. This would be discussed in possible revised draft.

page 4, line 4-10: These results are easily expected before the experiments. The results are like that, simply because of the assumptions given to the mass balance.
In above replies I have hopefully justified why such a  simple mass balance function is enough  to investigate some specific questions related to the recent thinning rates in Himalayan glaciers with and without a supraglacial debris-cover.

---

## Referee Comment (RC2) · Anonymous Referee #2 · 12 Sep 2016

This paper uses a simplified flowline model to assess the impact of debris cover on debris-covered and debris free glaciers. The underlying concepts the paper aims to test are relevant and important. However, I think that the way in which debris included creates a circular argument. Areas with 'debris' cannot thin below a threshold, but this threshold covers a large portion of the ablation area in the model glacier. The author then uses this result to show that the debris covered area has not thinned, whereas the ice free glacier, which does not have this limit, does not thin. To me, this does not tell us about debris cover, but uses an arbitrary threshold to stop thinning at a certain point on one glaciers, but not on another. This debris parameterisation is fundamental to the paper. If my understanding is correct, then it is fundamentally flawed and circular and does not give us any information about the impact of debris on glacier melt rates. Page 1 Line2: Where as thin debris cover is expected to accelerate melt, due to its low

albedo. Line 5: in >on Line 6: The sentence starting 'Subsequently. . ..' Is hard to follow. Express more clearly. Line 7: I find this sentence hard to understand (starting 'Time evolution. . ..') Line 13: Outline the impacts of thin debris cover on ice loss. Line 15: .. length change, and formation of.. Line 18: This task is made more difficult by our limited understanding of. . . Page 2 Line 2: Why then should. . ... Line 5: get compensated> be compensated for. Line 8: Very briefly outline what these are. Line 11: pointed out> highlighted. Pointed out is colloquial. Line 12 debris-covered glaciers, but. Should be a comma not full stop. Line 27: steady state THE ice thickness profile. Page 3 Line 1: If I have understood correctly, the debris cover is applied by simply saturating ablation at -2 ma-1 over part of the terminus. This therefore seems like a very circular argument, as the value for this section cannot become less than -2 ma-1. It therefore cannot thin and this is then used in an argument to say that debris cover means that the glacier does not thin. The only thing that can change is the upper section, which does thin. To be, this is circular parametrisation and not an appropriate way to evaluate the impact of debris cover. Perhaps I have misunderstood this, but it needs to be explained more clearly. Also, why a value of -2 ma-1? Line 17: I don't follow this argument.
* * *

---

## Author Response (AR1)

I thank the anonymous referees for their critical review of the discussion paper and useful suggestions. Below I list my resplies to the comments of Anonymous Referee #1 and #2. The red lines are referee's comments and the corresponding replies are in black. The relevant changes in the revised manuscript are listed by referring to the corresponding page/line numbers within parenthesis.

**Authors response to review comments of Anonymous Referee #1:**
* It is difficult to find the significance of the study. Glacier thinning occurs by a combination of the surface mass balance and the emergence velocity. Initial change in ice thickness is controlled by surface mass balance, and then affected by changes in glacier dynamics later. Response time of a debris-covered glacier is generally longer than that of debris-free glaciers. All these were frequently argued and well demonstrated in previous studies.Therefore, it is not surprising to see the results shown in Figure 2.

Undoubtedly glacier thinning is controlled by conservation of mass, a slow dynamics of ice and a fast changing mass-balance forcing. I do not claim to have introduced this ideas here in this paper for the first time. However, to the best of my knowledge, these basic principles were not applied in interpreting the recent large scale glacier thinning data from debris-covered and debris-free glaciers in the Himalaya (Kääb et al , 2012; Gardelle et al , 2012; Nuimura et al , 2012; Gardelle et al , 2013; Vincent et al, 2016), leading to the apparent and well-known puzzle that has been outlined in detail in the introduction section. Despite such a long list of well-known papers that have dealt with this issue, Vincent et al (2016) have recently stated: "This question of area-averaged melting rates over debris-covered or clean glacier ablation areas remains unanswered".  This shows that a clear understanding of this effect has been lacking in the present literature so far. This paper addresses the issue based on first principles.

If the effect has  already been clearly explained in some reference that is not known to me, I am ready to accept that the present contribution is redundant.

* Moreover, the model and experimental conditions are very simple (1D flow line model, simple ice dynamics and mass balance). Among others, this study neglects important aspects of a debris-covered glacier, which are listed in the introduction of the paper (line 19-20); time-evolution of the debris extent, variability of debris thickness, and highly dynamic supraglacial ponds and ice cliffs.

I apologise to the reviewer and the readers for not providing a  detailed justification of the simple model used in this paper. I thank the reviewer for pointing this weakness out. This is now discussed in the revised version (P2 L6, L9-13, L15-20).

The basic point here is that the relatively fast spatio-temporal variations of melt-rate due to the advecting ephemeral thermokarst features (ponds and cliffs) on the glacier surface and an inhomogeneous debris layer, in combination with a slow response of debris-covered glaciers, imply that long term avarage melt-rate is a rather well-defined quantity and that is what that controls the thinning dynamics at any given point, $x$, over deacal scale.  Moreover, as pointed out in the article, the present data suggest, the theromokarst feature play a relatively weaker role in terms of the total melt - at the level of  10-20% (Sakai et al, (2000); Reid and Brock, (2014)).

In addition, since the quoted thinning data are from a  large ensemble of glaciers, another level of averaging over such a large ensemble would get rid of the effects of specific details of the mass balance the individual glaciers.

Therefore, it is justified to use a simple (and thus tractable) average mass balance curve to investigate the question of large scale thinning rates in glaciers in the Himalaya. The specific meltcurve used here is motivated by data from Himalayan glaciers (Chhota Shigri, Hamtah, Dokriani and Chora Bari glaciers; eg Banerjee and Azam, 2015). A more complicated representative melt-curve would not change our basic results.

There is a possibility that climatic forcing may increase the average melt rate or may lead to higher abundance of ponds/cliffs (discussed later in the reply), and thus changing the mean melt-rates near the tongue. Given the lack of long term data, this effect is hard to quantify at present. The fact that there are number of debris-covered glaciers with large stagnant tongues in the Himalaya (Scherler et al , 2011), may be a pointer that this increase is not very significant in terms of its magnitude. The idealised mass balance used here, captures the formation of the stagnant tongue quite well.

The uppper elevation range of the thickly debris-covered region has been assumed to increase in our idealised debris-covered glacier model by the same amount as the ELA, to take care of the possible increase of debris covered area in a simple way.

 * In any case, the paper is too short to report complex behavior of debris-covered glaciers.

As explained above, the aim here is to investigate the specific question of decadal scale data of thinning rates of a large collection of debris-covered vs debris-free glaciers. I have argued that the model/paper is adequate for this specific purpose.

I do not intend to explain all aspects of the complex behaviour of debris covered glaciers. The complexities alluded to above, would only be relevant in answering more detailed questions like the pattern of thinning in a specific glacier and can be safely ignored in the present context.

Besides, the existing detailed models may not be capable of simulating the large ensemble of glaciers considered here. Most of these models require very high resolution baseline data related to glacier and climate, which may not be available.

* I list below specific comments on the manuscript.
page 1, line19-20: These are very important aspects, but completely neglected in the study.

I have already explained my view on this issue in the response detailed above.

* page 2, line 28: "vertical ablation" is odd. Do you mean "surface ablation"?

It is corrected.

* page 3, line 3-4: "mass balance shape remains the same" » This is a very crude assumption because the debris layer thickens and lakes are formed.

A thickening debris layer would affect the mass balance values for sure. However, in the thickly debris covered parts of the glacier this effect would be relatively unimportant. This is evident from the known variation of melt rate under a debris layer (Ostrem curve) that shows smaller decrease in melt-rate in the thick debris limit (more than about ~10 cm). (eg Juen et al, The Cryosphere, 8, 377–386, 2014). And, the possible increase in debris-extent is included in an empirical manner by moving up the saturated portion of the melt-rate curve as ELA goes up (expected in case of melt-out debris).

On the other hand, supra glacial lakes, as pointed out before, only contribute ~10-20% of the total melt (Sakai et al, (2000); Reid and Brock, (2014)) for specific glaciers studied. Also large-scale

studies (eg Gardelle et al, Global and Planetary Change, Elsevier, 2011, 75 (1-2), pp.47-55) reveal that the *supraglacial* lake area is typically only a fraction of a percent of the total glacierised area in the region, and that the total supraglacial lake area is growing at a rate of a few 10's of percent or less per decade (with large uncertainties in the estimates). So the net effective lowering of melt rates due these possibly increasing supraglacial lakes can be ignored in the first approximation.

These discussions is incorporated in the revised draft (P2 L15-20 ).

* page 3, line 13-14: The result is not "interesting" if "this is an artifact".
I agree with the reviewer and appropriate changes have been made.

* page 3, line 25: What is the unit of the mass balance gradient?
Unit is specified now.

* page 3, line 3: Why 30 m (not 50 m)?
A change of 50m at the rate of 1m every five year, requires a total of 250 years, stretching the time axis of the figure 2 – that is why we had truncated it at 150 years ie a total change of 30 years. This has been mentioned in possible revised draft (P4 L17-19).

* page 4, line 4-10: These results are easily expected before the experiments. The results are like that, simply because of the assumptions given to the mass balance.
In above replies I have hopefully justified why such a  simple mass balance function is enough  to investigate the important specific issue of the recent thinning rates in Himalayan glaciers with and without a supraglacial debris-cover.

**Authors response to review comments  of Anonymous Referee #2:**

* I think that the way in which debris included creates a circular argument. Areas with 'debris' cannot thin below a threshold, but this threshold covers a large portion of the ablation area in the model glacier. The author then uses this result to show that the debris covered area has not thinned, whereas the ice free glacier, which does not have this limit, does not thin. To me, this does not tell us about debris cover, but uses an arbitrary threshold to stop thinning at a certain point on one glaciers, but not on another. This debris parameterisation is fundamental to the paper. If my understanding is correct, then it is fundamentally flawed and circular and does not give us any information about the impact of debris on glacier melt rates.

 I believe the claim that the paper uses a circular argument is incorrect. Here are my arguments:

1) Our modeled thinning rates are in the range ~0.2-0.3 mwe/yr.   The assumed threshold value and corresponding mean melt rate is much larger.

2) The mean melt-rate in principle does set an upper bound on thinning rate, but that bound is irrelevant here. Immediately after a step change of ELA, the debris covered parts do not thin. This is not becasue of a low melt rate there, but because the mass balance curve is *flat* there. The actual value of the threshold does not matter - The same effect would be seen if the treshold value was larger, as is clear from figure 1b (Though the time scale of stagnation may be smaller in that case).

3) As explained in the paper, the interplay of the changes in mass balance forcing and a slowly evolving flux-divergence profile controls the net thinning of any glacier, as opposed to the melt rate being the only controling factor. That is why one has higher/lower/similar mean thinning rate in

debris-covered glaciers as compared to their debris-free counterparts, depending on the stage of response and the rate of mass-balance change (eg, our numerical results clearly show that debris-covered glaciers thin at a faster rate during the later stages of the response (fig 2, expt 1,2)).

4) The "threshold" on melt rate in the thickly debris covered part is supported by glaciological mass balance data from Himalayan glaciers (eg Banerjee and Azam, 2015) and is not an arbitrary imaginary construct (as explained in the revised draft (P3 L5-7)). The threshold does exist and also its exact magnitude is not important for the effects described here (as explained in point 2).

5) That the model describes and explains the stagnant debris-covered glaciers commonly observed in the Himalaya that have formed in response to a warming climate (Scherler et al, 2011; Banerjee and Shankar, 2013) indicates that it describes debris covered glaciers reasonably well.

* Page 1 Line2: Where as thin debris cover is expected to accelerate melt, due to its low albedo
I prefer to leave it out of the abstract. The welknown albedo effect of a thin debris layer does not seem to be visible in the measured mass balance profile of Himalayan glaciers (Banerjee and Azam, 2015).  It is likely that the thin debris extent is small compared to that of the thickly debris region. May be the accelerated melt in the thin debris region contributes a large melt-out debris flux, leading to quick thickening of the debris layer.
I have included this line in the introduciton (P1 L14)

* Line 5: in >on
Line 6: The sentence starting 'Subsequently. . ..' Is hard to follow.
Express more clearly.
Line 7: I find this sentence hard to understand (starting 'Time evolution. . ..')

Based on above suggestions, the abstract has been rewritten.

* Line 13: Outline the impacts of thin debris cover on ice loss.
Line 15: .. length change, and formation of..
Line 18: This task is made more difficult by our limited understanding of. . .
Page 2 Line 2: Why then should. . ..
Line 5: get compensated> be compensated for.
Line 8: Very briefly outline what these are.
Line 11: pointed out> highlighted. Pointed out is colloquial.
Line 12 debris-covered glaciers, but. Should be a comma not full stop.
Line 27: steady state THE ice thickness profile.

All above suggestions/corrections have been incorporated in the revised draft.

* Page 3 Line 1: If I have understood correctly, the debris cover is applied by simply saturating ablation at -2 ma-1 over part of the terminus. This therefore seems like a very circular argument, as the value for this section cannot become less than -2 ma-1. It therefore cannot thin and this is then used in an argument to say that debris cover means that the glacier does not thin. The only thing that can change is the upper section, which does thin. To be, this is circular parametrisation and not an appropriate way to evaluate the impact of debris cover. Perhaps I have misunderstood this, but it needs to be explained more clearly. Also, why a value of -2 ma-1?

We have have outlined our reply to this objection in the beginnning  of the section.

* Line 17: I don't follow this argument.

We have added some clarifications to make the point clearer. (P3 L34-35, P4 L1-6)

[revised manuscript text omitted]

---

## Author Response (AR2)

Revisions suggested by Anonymous Referee #3 (Report #2):

More substantial points:

a) In general the writing (English language, punctuation…) needs to be improved (quite a few awkward formulations and wrong order of terms in sentences) and minor editing issues (such as for example, singular/plural, wrong spaces, brackets and commas around citations etc.) need to be eliminated by carefully checking and proof reading the manuscript in a further revision. As there were too many such issues I did not mark them all in the detailed comments below.

I have tried to rectify the errors to the best of my ability.

b) (p. 4 first paragraph): I struggle to follow the line of argument on line 3 and 4. On what basis the authors comes to the conclusion that why in a later stage thinning rate would in A become smaller than in glacier B and sentence. Is the reason due to the delayed response of B whereas A already is adjusted to the changed climate. I can very well see this conclusion from the later modelling but at this stage it seems not so obvious to me.
Should be clarified and better explained.

I have rewritten this paragraph following the suggestions by the reviewer.

c) Figure 1 c and d (and e and f): Figure 1 is very instructive, but unfortunately it is difficult to see the temporal evolution of thing rates or dF/dx which is however very crucial to understand what is going on. One should be able to see which lines are at which time step. A reader who is experienced with such model output can probably read it right but not the general reader.
Either label the different blue lines with the model years or maybe visually easier color them following a easy to read color code (rainbow colors that change with time).

I have made the changes in the figure as suggested by the reviewer.

d) If I understand right a crucial point in that the thinning rates from debris covered tongues get similar or higher than on debris-free tongues is the delay in dynamic response of the debris covered tongues. Or in other words the ice-free tongues have already adjusted when the debris covered parts are approaching their highest thinning rates (or do I get this wrong?). This point of delayed response should in my view be made more explicit in paragraph '3.1.2. time evolution of thinning rates' at it seems it is all about timing.

I have included the statement: "the emergence velocity profile in the lower ablation zone of the debris-covered glacier shows a delayed response (figure 1f) leading to a lower value of the glacier-averaged initial thinning rate"

e) More a note to further support this study: Often elevation change assessment are focussed on lower parts of glaciers (ablation areas, as not too steep), but the suggested dynamic effect on average thinning rates would probably be even more amplified and clearer if focussed only on for example the ablation area. This would further support the conclusions of this paper.

I did discuss that in section 3.1.2: "While we have considered the glacier-wide thinning rate, the same conclusions are obtained if one compares only the lower part of the two glaciers  as they are identical  in their upper parts". However, we prefer to compare the glacier-averaged thinning rate that is equivalent to the net specific blance – an well-accepted fundamental observable for glaciers.

Detailed minor comments and editing issues:

p. 1 line 12: I would say '…in its DYNAMIC response…' to flag more the DYNAMIC aspect.
p. 1 line 12: here only modelling studies are cited but surely there are 'other' studies that considered 'dynamic effects' before.
p.1 line 23: '…has emerged FROM the large…'
p. 1 line 25 and p 2 line 1: I would reformulate tis to '…of supraglacial debris-cover and may seem counterintuitive.'
p. 2 line 7: '…melting FROM thermokarst PROCESSES, namely'
p. 2 line 33: for clarification I would add a 'initially' before 'just the difference
p. 3 line one: something wrong in the sentence at end of line: '…over a time scale THAT is short COMPARED to the …'?
p. 3 line 7: singular 'glacier'
p. 3 line 10 'mass-balance' (double 'ss')
p. 3 line 13 : profile should be in plural: '…mass-balance profiles…'
p. 3 line 14: '…and only CHANGES only by the shift in ELA, and no reference needed here.

All of the above suggestions have been accepted.

p. 3 line 18: I am sure some readers will not agree with the point that changes in ice cliffs/ponds are NOT IMPORTANT. The point is that in this study it makes sense to explicitly exclude it as it wants to see what the 'dynamic' effect is. Thus, in my view there is no need to say the ice cliffs etc are not important or the effect is small.

I have rewritten the section emphasising that the ice-cliffs/ponds are being neglected only as a first approximation to focuss oon the effects of the flux dynamics.

p. 3 line 9: I think even without 'fast' advection this simple mass-balance profile is justified.

I agree with the reviewer. However, this discussion was included in the previous draft in response to a reviewer's comments and is left unchanged here to clarify that the mass-balance profile used affords a reasonable description of debris-covered glaciers .

p. 3 line 21: I would add 'initially' again between '..responds' and 'with a…'
p. 4 line 15, 16: YEARS should be in plural on both lines
p.4 line 17: i assume these decadal rates refer to the first (few) decade and later decline.
p. 4 line 25: I assume these is the '…initial AVERAGE thinning rate (averaged along glacier)
p. 5 line 3: '… profile of THE thinning rate gets …'
Fig 1, caption: make clear that this figure refers to experiment 1.

All of the above suggestions have been accepted in this version.

[revised manuscript text omitted]